# Multi-scale analysis of the Asian Monsoon change in the

- 2 last millennium
- 3

## 4 TingGui Jiang<sup>1,2,3</sup>, JiaJia Lin<sup>4</sup>, ZhenShan Lin<sup>1,2,3,\*</sup>, and YuXia Li<sup>1,2,3</sup>

- <sup>5</sup> <sup>1</sup>College of Geography Science, Nanjing Normal University, Nanjing, China
- 6 <sup>2</sup>Jiangsu Key Laboratory of Environmental Change and Ecological Construction, Nanjing,
- 7 China
- <sup>3</sup>Jiangsu Center for Collaborative Innovation in Geographical Information Resource
  Development and Application, Nanjing, China
- <sup>4</sup>Department of Earth and Environmental Sciences, University of Illinois at Chicago, Chicago,
- 11 IL, USA
- 12 Correspondence to: ZhenShan Lin (linzhenshan@njnu.edu.cn)
- 13

#### 14 Abstract

The change rule and the driving mechanism of the Asian monsoon (AM) is widely concerned. 15 In this paper, we have decomposed the Huangye cave  $\delta^{18}O_R$  datasets (proxy for AM) into 16 variations at different time scales with the Ensemble Empirical Mode Decomposition (EEMD) 17 method, obtained the main cycles of the Asian monsoon changes and drought and wet periods 18 in the last millennial history, analysed and predicted the precipitation trend at the local site in 19 the future. Meantime, we have also decomposed <sup>10</sup>Be (proxy for solar activity) and average 20 21 Northern Hemisphere Temperature (NHT) datasets with the EEMD method. The relationship among the components of  $\delta^{18}O_R$ , <sup>10</sup>Be and NHT shows that NHT as well as solar activity has 22 23 the obvious driving effect on the AM at millennial scales, and NHT with the frequency and amplitude modulations can influence the intensity of the AM more directly and obviously than 24 25 the solar activity in the last 1000 years.

1

### 2 1 Introduction

The Asian Monsoon (AM) is an important part of the global monsoon and climate system, 3 the strength of which affects more than 60% population in the world (Zong et al., 2006; Lin et 4 al., 2012). The change of monsoon rainfall will cause extreme hydrological events, such as 5 massive floods or serious droughts, and a large number of reductions of agricultural products, 6 7 thus influence the migration of the population and production and living of the nation (Yang et 8 al., 2005; Wang, 2008; Tan et al., 2011a). Many evidences show that there were abrupt changes 9 and periodic fluctuations in Asian monsoon in the past 1000 year (Sinha et al., 2011; Tan et al., 10 2011a; Tan et al., 2011b; Zhao et al., 2015), and the climatic changes had played an important 11 role in Chinese societal evolution (Zhang et al., 2008; Tan et al., 2011a). The oxygen isotopic value of the cave stalagmite is one of the most important indicators of the ancient climate, which 12 13 contains abundant information of climate change, and can be used to indicate intensity 14 variability of the Asian monsoon (Wang et al., 2005; Wang et al., 2008; Zhang et al., 2008; Tan et al., 2011a; Tan et al., 2011b; Liu et al., 2012). There are many foctors affecting the Asian 15 16 Monsoon, including solar activity, the Northern Hemisphere Temperature (NHT), El Nino 17 /Southern Oscillation (ENSO), volcanic activities, and greenhouse gases (An et al., 2000; Wang 18 et al., 2005; Wang et al., 2008; Zhang et al., 2008; Tan et al., 2011a; Tan et al., 2011b; Liu et al., 19 2012; Jiang et al., 2013; Anderson et al., 2002; Mann et al., 2005; Liu and Yanai, 2002; Wang 20 et al., 2010).

The Asian Monsoon exhibits all kinds of cyclical variabilities, and different cycles may be 21 22 driven by different factors. A number of studies reveal the solar activity has important 23 influencs on the Asian Monsoon change (Fleitmann et al., 2003; Dykoski et al., 2005; Tan et al., 2008; Wang et al., 2008; Liu et al., 2012). The ralationship between the AM and NHT is 24 also largely expounded (Zhang et al., 2008; Tan et al., 2011a; Tan et al., 2011b; Yang et al., 25 2014). Previous studies are usually on single scales, relationship among the indexes are 26 mostly identified by direct comparison (Wang et al., 2005; Tan et al., 2008; Sinha et al., 2011; 27 Zhao et al., 2015), the cycles are analysed by conventional method such as power spectral 28 analysis (Tan et al., 2009; Man and Zhou, 2011). So it is difficult to reveal the shifts at 29

different time scales and how the variations of different time scales repond to different driving

- forces.
- A good time-frequency analysis method, Empirical Mode Decomposition (EMD) has been
- proposed by Huang et al. in 1998. And in 2009 Wu and Huang presented its modified version,
- Ensemble Empirical Mode Decomposition (EEMD). This method is very suitable for non-
- linear and non-stationary time series. The EEMD method can quickly decompose the signal
- into different periodic fluctuations at frequencies from high to low, and the last residue is the
- trend component of data sequence. The fluctuation of different cycle is defined as the Intrinsic
- Mode Function (IMF). The IMF fluctuation component has significant characteristic of
- graded wave, and the trend component is the monotonic function or mean function. The
- fluctuation of IMF component is steady and nonlinear, and each component means a physical
- fluctuation variability with a different characteristic time scale. (Lin et al., 2006; Lin et al.,
- 2007; Liu et al., 2012).

In this paper, we will apply the EEMD method to reveal the periodiciyies of AM and the
possible link between AM with solar activity and NHT.

16

#### 17 2. EEMD method and data

#### 18 2.1 EEMD method

Ensemble empirical mode decomposition (EEMD) (Wu and Huang, 2009) is an enhancement 19 20 of the empirical mode decomposition (EMD), which is an empirical but highly efficient and adaptive method for processing non-linear and non-stationary time series (Huang et al., 1998; 21 Wu and Huang, 2004). The specific process of EMD is as follows: firstly all the extrema of the 22 original time series are identified, all the local maxima are connected by a cubic spline line as 23 24 the upper envelope. Repeat the procedure for the local minima to produce the lower envelope. 25 All the data must lies between the upper and lower envelopes. Their mean is designated as  $m_l$ , 26 and the difference between the data and  $m_1$  is the first component,  $h_1$ , i.e.

27 
$$X(t) - m_1(t) = h_1(t)$$
 (1)

28 Repeat the procedure, that is

29 
$$h_1(t) - m_2(t) = h_2(t)$$

3

(2)

- 1 And further iterations of this sifting process should be executed until standard deviation can
- 2 be set between 0.2 and 0.3.

3 
$$SD = \sum_{t=0}^{T} \left[ \frac{|(h_{i-1}(t) - h_i(t))|^2}{h_{i-1}^2(t)} \right]$$
 (3)

Hence, the first IMF component  $c_1$  comes from the data,  $c_1 = h_i(t)$ , and  $c_1$  should contain the finest scale or the shortest period component of the signal. After that,  $r_1$  is designated as the rest of the data by  $r_1 = X(t) - c_1$ . Since  $r_1$ , still contains information of longer period components, it is treated as the new data and subjected to the same sifting process as described above. This procedure is repeated and the result is

9 
$$r_2 = r_1 - c_2, \dots, r_{n-1} = r_n - c_n$$
 (4)

The sifting process can be stopped until  $r_n$  becomes a monotonic function from which no more IMF can be extracted. So

12 
$$X(t) = \sum_{i=1}^{n} c_i + r_n$$
 (5)

or

14 
$$X(t) = \sum_{i=1}^{n} IMF_i(t) + r_n(t)$$
 (6)

However, EMD has a potential mode-mixing problem, which can render EMD unable to represent the characteristics of the original data (Wu and Huang, 2009). To overcome the drawback, a new Ensemble Empirical Mode Decomposition (EEMD) is presented by Wu and Huang in 2009. A white noise series will be added to the targeted data in the EEMD method, the data with added white noise is decomposed into IMFs, then step 1 and step 2 are repeated again and again, but with different white noise series each time. Ensemble means of the corresponding IMFs of the decomposed signals are calculated by

22 
$$c_i(t) = \frac{1}{M} \sum_{m=1}^{M} c_i^m(t)$$
 (7)

At last we will obtain the ensemble means of corresponding IMFs which indicate different periodic fluctuations at frequencies from high to low and the last residue that is the trend component of data sequence as the final result.

#### 26 **2.2 Data source**

The oxgyen isotope ratio ( $\delta^{18}$ O) of the stalagmite from cave can reflect change of  $\delta^{18}$ O of meteoric precipitation at the site, which in turn relate to the amount of precipitation (Wang et al., 2005; Dykoski et al., 2005; Zhang et al., 2008; Tan et al., 2011a; Tan et al., 2011b). The

 $\delta^{18}$ O values of three stalagmites HY1, HY2 and HY3 were tested and were combined to a 1 reconstructed Huangye stalagmite  $\delta^{18}O_R$  record between AD 138 to 2003, with a resolution of 2 2-7 years, which showed substantial variability in the monsoon precipitation (Tan et al., 2011a). 3 The Huangye Cave (33°35'N, 105°07'E, 1650 m above sea level at its entrance) is located about 4 20 km northeast of Wudou County, eastern Gansu Province, China (Zhang et al., 2005; Tan et 5 al., 2011a). This region belongs the semi-humid region of northern China, most of the rainfall 6 7 (  $\sim 80\%$ , for the period AD 1951–2003) occurs during the summer monsoon months (May– September) (Tan et al., 2011a). The AM stengthens with its rain belt moving northwards, 8 bringing more rainfall at the site. The Huangye Cave stalagmite  $\delta^{18}O_R$  values reflect the 9 monsoon precipitation changes, with lower  $\delta^{18}O_R$  values representing higher precipitation and 10 11 vice versa (Tan et al., 2011a).

In this article, we selected Huangye Cave  $\delta^{18}O_R$  series at AD 1000-2002 to analyze Asian Monsoon variabilities in the past 1000 years, and to compare with other proxy. Before using the EEMD method, we applied cubic spline function with 2 years step interpolation to dataseries of  $\delta^{18}O_R$  in the period of AD 1000-2002 for equal spacing.

Beryllium-10 (<sup>10</sup>Be) and carbon-14 (<sup>14</sup>C) record are considered most reliable proxies of change in solar activity (Yiou et al., 1997; Stuiver et al., 1998; Bard et al.,2000; Muscheler et al., 2007). But many proxy data of <sup>10</sup>Be and <sup>14</sup>C have low temporal resolution (Muscheler et al., 2007; Delaygue and Bard, 2011). We adopted <sup>10</sup>Be with annual resolution from Crowley as the proxy of solar activity over the past 1000 years, the data seris were compounded by the <sup>10</sup>Be measurements from Antarctica (Bard et al. 1997), spliced into the Lean et al. (1995) record (Crowley, 2000).

Crowley had also reconstructed Northern Hemisphere Temperature from the mean annual temperature of Mann et al. and of Crowley and Lowery (Crowley, 2000). The time length of Northern Hemisphere Temperature is from 1000 to 1965 ,and is shorter than  $\delta^{18}O_R$  and  $^{10}Be$ (Fig. 1).

Fig. 1

28

27

29 **3.** EEMD analysis of the  $\delta^{18}O_R$ 

1 The EEMD method was utilized to extract the intrinsic cycles of the time series of  $\delta^{18}O_R$ . 2 The stalagmites  $\delta^{18}O_R$  datasets of the Huangye Cave from AD1000 to and 2002 was 3 decomposed into seven IMF components and a residue as the trend (Fig. 2). Their cycles and 4 corresponding variance contribution rates are shown in Table 1.

> Fig. 2 Table 1

|  |  | ٠ |   |
|--|--|---|---|
|  |  |   |   |
|  |  |   |   |
|  |  | - | - |
|  |  |   |   |
|  |  |   |   |

6

The cycles of the IMF components of  $\delta^{18}O_R$  centre on 12.6, 24.1, 58.9yr, which are close to the cycles of 10.2yr, 25.4yr, 57.4yr in the spectral analysis (Fig. 3). The long cycles are seldom in the spectral analysis except for the cycles of 100.4yr and 154.5yr, and their confidence levels are lower than 90%. The cycles of 135.4yr which is strong in the EEMD method cannot be found in the spectral analysis.

12

### Fig. 3

In Table 1 the trend (Res) of  $\delta^{18}O_R$  has the largest variance contribution of 40.6%, IMF4 has the second largest variance contribution rate of 24.2%. When IMF4 is stacked to the residue(Fig. 4), the sum of the variance contribution rate will reach to 64.8%. The curve can approximately indicate the change rule of  $\delta^{18}O_R$ . In Figure 4, there are three significant troughs, AD 1125, AD 1300, and AD 1910. The higher  $\delta^{18}O_R$  value is, the drier the climate is, vice versa, The lower  $\delta^{18}O_R$  value is, the wetter the climate is.

19

### Fig. 4

20 So AD 1100-1150 is the wettest period, AD 1890-1930 is the second wettest period. It is mainly consistent with the conclusion drawn by Tan et al. that AD 1090-1140 and AD 1880-21 1920 are wetter periods. We also reveal AD 1290-1310 is wet period that Tan et al. didnot find. 22 23 Figure 4 also reveals seven peaks, AD 1220, AD 1370, AD 1550, AD 1660, AD 1740, AD 1825, and AD 2000. The values of the two peaks of AD 1370, AD 1550 are more than 0.3, the 24 values of the peaks of AD 1220, AD 1660, AD 1740, AD 1825 and AD 2000 are more than 0.2 25 and less than 0.3. So we have reached to the following conclusions: AD 1350-1390 and AD 26 1530-1570 are extremely drought periods; AD 1220±15, AD 1660±15, AD 1740±15, AD 27 1825 ±15 and AD 2000 ±15 are drought periods. These statements are not as same as conclusions 28 of Tan et al. that ~ AD 1350, AD 1610-1650 are relatively drought by the analysis of the 29 precipitation index. Our conclusions are more concrete and straightforward. 30

Figure 4 also indicates that the monsoon will strengthen gradually in the next decades or even
the next 200 years, at the same time the precipitation will also increase, and it will be the next
wettest period in ~ AD 2180±30.

4

#### 5 4. Relations among AM, solar activity, and NHT at different time scales

Figure 5 shows that the <sup>10</sup>Be signal and Northern Hemisphere Temperature time series are
decomposed into IMF components and trend (Res) with the EEMD method, the two variations
(signals) both contain eight quasi-period oscillation on various time scales and a trend. Table 2
shows periodicities and their variance contribution rates.

Fig. 5

Table 2

- 10
- 11

As shown in Fig. 1, the dataset of <sup>10</sup>Be in the period of AD 1000-1700 is level and smooth, 12 it lead to the fluctuation of IMF1, IMF2 and IMF3 of <sup>10</sup>Be is indistinctive in the period (Fig. 13 5a). The mean cycle of IMF1 of <sup>10</sup>Be is 3.5yr in AD 1000-1998, but the mean cycle in AD 1730-14 1998 is 4.1yr. Similarly, the cycles of IMF2 and IMF3 of <sup>10</sup>Be in AD 1000-1998 is 7.8yr and 15 20.5yr, the cycle in AD 1730-1998 is 11.2yr and 22.6yr. <sup>10</sup>Be shows three cycles of 124.8yr, 16 11.1yr and 9.9yr by the power spectral analysis (Fig. 6a). The cycle of 11.1yr is close to the 17 cycle of 11.2yr of IMF3 in AD 1730-1998 by the EEMD, and the cycle of 124.8yr in the spectral 18 analysis can also be found as IMF5 with the cycle of 121yr. But the other cycles cannot be 19 20 found with exceeding 80% confidence level in the spectral analysis. The Fig. 6b has shown many small time-scale cycles of NHT in power spectrum, doesnot reveal the apparent cycle of 21 more than 35.8yr, small time-scale cycles such as 7.8yr and 26.8yr are closed to the cycles of 22 8.3yr and 24.7yr by the EEMD. 23

24

### Fig. 6

As shown in Table 3, the correlation coefficients of  $\delta^{18}O_R$  with NHT are higher than with <sup>10</sup>Be. Except for IMF5, the correlation coefficients between  $\delta^{18}O_R$  and NHT are all negative, and the cycles are longer, the the correlation coefficients are greater. To reveal possible driving modulation at long time scales, the IMF7 and residue of  $\delta^{18}O_R$  have been compared with the IMF7 and residue of <sup>10</sup>Be, NHT (Fig. 7). In the trend, the residue of NHT is coincide with the

residue of <sup>10</sup>Be, meantime they are both in inverse phase with the residue of  $\delta^{18}O_R$ . The trend 1 of  $\delta^{18}O_R$  has better inverse corresponding relation with the trend of NHT than the trend of  $^{10}Be$ . 2 On the whole, the correlation coefficient between the trend of  $\delta^{18}O_R$  and the trend of  ${}^{10}Be$  is 3 only -0.1057 (p<0.05), but in AD 1200-1998 the correlation coefficient reaches to -0.4827 4 (p<0.01), shows a more clear correlation. The IMF7 of NHT is coincide with the IMF7 of  $^{10}$ Be, 5 although there are partly postpone in the period of AD 1200 to 1600. And the IMF7 of NHT 6 has a better correlation with the IMF7 of  $\delta^{18}O_R$  than the IMF7 of  $^{10}Be$ . Therefore, at long time 7 scales, the solar activity and NHT both have the obvious driving effect on the Asian monsoon. 8 Solar activity is more violent, the average temperature of the northern hemisphere is higher, the 9 Asian monsoon is more stronger. As shown in Fig. 7 (a) and (b), NHT has the more prominent 10 negative correlation with  $\delta^{18}O_R$  than the solar activity at ~900yr time scale and the trend. The 11 average northern hemisphere temperature seems to have a more direct drive for the Asian 12 13 monsoon in the last millennium.

- 14
- 15
- Because the high frequency components has some uncertainty and bias, we had analysed the 16 envelopes of high frequency components of  $\delta^{18}O_R$ . The envelope of the IMF1 of  $\delta^{18}O_R$  has the 17 mean period of 51.3yr, the envelope of the IMF2 and IMF3 of  $\delta^{18}O_R$  has respectively the mean 18 19 period of 102yr and 213yr. These cycles are close to the cycles of IMF4, IMF5 and IMF6 of 20 NHT, but they have more gap with the cycles of IMF4, IMF5 and IMF6 of <sup>10</sup>Be. Figure 8 shows the comparisons between the envelopes of IMFs of  $\delta^{18}O_R$  and IMFs of NHT. In spite of some 21 malposition, the crests of the envelope of the IMF1 of  $\delta^{18}O_R$  coincide with the troughs of the 22 IMF4 of NHT and vice versa (Fig. 8a). Similarly, the envelopes of the IMF2 and IMF3 of  $\delta^{18}O_R$ 23 are also in inverse phase with the IMF5 and IMF6 of NHT in large part (Fig. 8b, Fig. 8c). So 24 Ansian monsoon may be driven by the frequency and amplitude modulations of NHT. 25

Table 3

Fig. 7

26 27

# 28 **5. Conclusions**

AM owns the cycles of 12.6yr, 24.1yr, 58.9yr, 135.4yr, 220.6yr, 718yr, 818yr from Huangye

Fig. 8

Cave stalagmites  $\delta^{18}O_R$  datasets by EEMD method. The fitting of the high-weight variance 1 contribution rate components of  $\delta^{18}O_R$  reveals it is the wettest period in AD 1100-1150, the 2 second wettest period is in AD 1890-1930. On the other hand, AD 1350-1390, AD 1530-1570 3 are extremely drought periods. Meantime we predict the Asian monsoon is strengthening 4 gradually and the Asian monsoon rainfall is increasing gradually in the next several decades or 5 even the next 200 years, in ~ AD 2180 ±30 the local climate will reach to the next wettest period. 6 The cycles of  $\delta^{18}O_R$  are similar to the cycles of  $^{10}Be$  and NHT at most time-scales by the 7 EEMD, which hints that there are possible internal responses between AM with solar activity 8 and NHT. With further analysis, we found that, at long-term scale, NHT as well as solar activity 9 has the obvious driving effect on the Asian monsoon. Solar activity is more violent, the average 10 11 Northern Hemisphere Temperature is higher, the Asian monsoon is more stronger and brings the more precipitation in the locality. The conclusion is similar to the many previous views 12 (Wang et al., 2005; Zhang et al., 2008; Tan et al., 2011a; Tan et al., 2011b). Our study further 13 14 reveals that the northern hemisphere temperature can influence the intensity of the Asian monsoon more directly than the solar activity in the last 1000 years. On the other hand, the 15 northern hemisphere temperature also influences the Asian monsoon by amplitude modulations 16 17 of shorter time scales. So Ansian monsoon may be driven by the frequency and amplitude 18 modulations of NHT.

19

Acknowledgements. This research has been supported by National Natural Science Foundation
of China (No. 31470519), Natural Science Foundation of Jiangsu Province (BK20131399) and
funded by the Priority Academic Program Development of Jiangsu Higher Education
Institutions. A lot of thanks are given to L. Tan and T.J. Crowley for their providing datasets.
We also thank the anonymous reviewers for helpful suggestions.

25

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

- Holocene forcing of the Indian Monsoon recorded in a stalagmite from southern Oman,