# Peer review of "Multi-scale analysis of the Asian Monsoon change in the"

_Nonlinear Processes in Geophysics, 2015_

## Referee Comment (RC1) · Anonymous Referee #1 · 7 May 2016

This paper used the EEMD method to analysis the AM variations in the last 1000 years. It gave a visual results of different cycles of AM in the last 1000 years, as well as their relationship with solar activity and NHT. It is an extent work of Tan et al. (2011a, b). Before it could be accepted, some major revisions are needed. 1. First of all, the authors concluded that the NHT can influence the AM more directly than solar activity, based on their statistics analysis results. The climate system is complicated. When discussing the driving forces, they need to discuss it from the aspect of climate dynamics, but not just from correlation coefficient aspect. Many factors may influence the correlation coefficient, such as age uncertainties. Even if two variances are significantly correlated, it doesnot mean one drive the other. They might be no connects. Tan et al. (2011 CP) have given a detailed discussion among solar activity, temperature, AM and precipitation changes. The author may want to refer. 2. The authors concluded that "we

predict the Asian monsoon is strengthening gradually and the Asian monsoon rainfall is increasing gradually in the next several decades or even the next 200 years, in $\sim$ AD 2180$\pm$30 the local climate will reach to the next wettest period". This is one of the main conclusion of the paper. However, neither it was shown in Figure 4, nor it was detailed discussed in the paper. 3. In page 6, the drought periods deduced from EEMD result are quite similar with Tan et al.' result (Tan et al., 2011a). For example, the drought occurred in $\sim$1350 AD and 1610-1650 AD were clearly shown in the abstract of Tan et al. (2011a). It is understandable, because they use the data of Tan et al. (2011a). 4. They can't deduce 718yr and 818yr cycles from a 1000-years long record. 5. I think the English of the paper need to be polished further. In addition, I also have some special comments: 1. Page4, line27-29: the speleothem d18O cant reflect rainfall amount in northern China(Zhang et al., 2008; Tan et al., 2011a; Tan et al., 2011b). ), but not in the region of Dongge Cave (Wang et al., 2005; Dykoski et al., 2005). Wang et al. and Dykoski et al. didn't claim that. 2. Page9, line 17: "Ansian monsoon" should be "Asian monsoon".

---

## Referee Comment (RC2) · Anonymous Referee #2 · 12 May 2016

This paper applies a Ensemble Empirical Mode Decomposition (EEMD) method and spectral analysis to paleoclimate records (isotopes from stalagmites in China, solar activity proxy and northern hemisphere mean temperature). The authors discuss the potential links between those variables.

Major comments: The methodology part (EEMD) is a repeat of already published material (Wu and Huang), but the presentation made by the authors is quite obscure, with ill-defined notations and no real effort at synthesizing the methodology and expressing its crucial elements.

The authors mainly repeat the description of the Huangye stalagmite data from the original publication, but they do not really discuss (or seem to doubt) how the isotopic record is relevant for their conclusions. The papers of Tan et al. mention correlations

between d18O and precipitation that are lower than 0.35 (from what I see in the data, this correlation is due to the trend of the last decades). Therefore, a lot of care should be used in interpreting and extrapolating such data.

There is no real discussion of what the cosmogenic isotope record means. The record used by the authors is a mix of several proxies, with different resolutions and temporal coverage. Therefore, the EEMD analysis might only tell something on the way the record was produced, and nothing about solar variability.

I am a bit surprised that the Fourier analysis of the authors does not find the same peaks as those found by Tan et al. (Palaeogeography, Palaeoclimatology, Palaeoecology 280, 432–439, 2009), with a very similar dataset and the same program for spectral analysis. Since a large part of the interpretation of data relies on those spectra, their instability casts some doubts on the results.

The spectral analysis results for the three records are very different. The authors seem to manipulate data until they find something that seems coherent. I do not approve of such a procedure.

In order to be complete, the authors should do the same exercise with volcanic activity, aerosol forcings, etc. The attribution exercise that is done here is very partial.

The authors make a prediction (p. 9, l. 7): "Meantime we predict the Asian monsoon is strengthening gradually and the Asian monsoon rainfall is increasing gradually in the next several decades or even the next 200 years, in $\sim$ AD 2180$\pm$30 the local climate will reach to the next wettest period." This cannot be serious. The data do not extend into the 21st century. Looking at the time series, I cannot see any hint temperature increases in the 21st century. And, this is were the correlation of $\sim$0.3 between precipitation and d18O might mean that precipitation reconstruction fails.

Minor comment: The English is very poor. The manuscript would have needed a cross check from a native English speaker.

I do not see how EEMD would give better or more insightful results than a wavelet decomposition (the ups and downs of the analysed signal are very symmetric).

The authors should be aware that Tom Crowley passed away in May 2014. Thanking him in the acknowledgments is rather strange.

Conclusion: For all the major comments, I am sorry to recommend a rejection of the paper.

---

## Author Comment (AC1) · 31 May 2016

Authors' Response to Anonymous Referee #1

We thank the referee for his (her) valuable comments and suggestions. Following is our response.

Referee's comment:
1. First of all, the authors concluded that the NHT can influence the AM more directly than solar activity, based on their statistics analysis results. The climate system is complicated. When discussing the driving forces, they need to discuss it from the aspect of climate dynamics, but not just from correlation coefficient aspect. Many factors may influence the correlation coefficient, such as age uncertainties. Even if two variances are significantly correlated, it doesnot mean one drive the other. They might be no connects. Tan et al. (2011 CP) have given a detailed discussion among solar activity, temperature, AM and precipitation changes. The author may want to refer.

Authors' Response:

Thanks for your comment. Indeed, even if two variances are significantly correlated, it doesnot mean one drive the other. But because the solar activity and NHT can influence the AM, many researchers infer the driving forces by the correlation coefficient. For example, Zhang Pingzhong et al. said: "The AM also has correlations to solar irradiance as inferred from $^{14}$C and $^{10}$Be records (22) [correlation coefficient (r)= –0.33, n= 345 data points for the past millennium, Fig. 2C and fig. S8]. These observations support the idea that solar forcing played a role in driving AM changes during the past two millennia."(Science, 322, 2008, Page941, line34-41).

But the inference is not particularly reliable.

For the clarity, We have changed the sentences as follows.

Page 8 Line 7-10:

Therefore, it suggests that the variations of the Asian monsoon have a close relationship with the solar activity at ~220, ~900yr time scales and trend, and the Asian monsoon have the obvious correlation to the average temperature of the northern hemisphere at ~60, ~120, ~900yr time scales and trend.

Page 9 Line 7-18:

The cycles of $\delta^{18}O_R$ are similar to the cycles of $^{10}$Be and NHT at most time-scales by the EEMD, which hints that there are possible internal responses between AM with solar activity and NHT. With further analysis, we found that the Asian monsoon has a close relationship with the solar activity at ~220, ~900yr time scales and trend, the Asian monsoon has the obvious correlation to the average temperature of the northern hemisphere at ~60, ~120, ~900yr time scales and trend. The Correlation coefficients of the Asian monsoon and NHT are so small at ~10 and ~24yr, it seems that there is no direct relation between them at the two time scales. Howerer, the variation intensity of the Asian monsoon at the two time scales is amplitude modulated by NHT at ~60 and ~120yr. Meantime, AM intensity at ~60yr is also amplitude modulated by NHT at ~220yr. So, AM variation may be closer relation with NHT than with the solar activity in the last 1000 years. It may be a pssible mechanism that AM can be driven at the long time scales by the solar activity, at the same time AM may be driven by the frequency and amplitude modulations of NHT in the last 1000 years.

The main work of Tan et al. (2011 CP) is to get the synthesized precipitation index record by selected four proxy records of precipitation, compare the synthesized precipitation record with the local BQ and DL temperature records, and "suggests warm-humid/cool-dry climate pattern in north central China during the last 1800 years". Only in 3.3 (Page689), Tan et al. compared synthesized precipitation record and the IAPO record with the solar activity records by direct comparision and spectral analysis periodicities ( without the figure of spectral analysis).

In the paper, we analysed the correlation of the corresponding IMF components of the original data by EEMD decomposition, in order to indicate the specific responds of AM to solar activity and NHT on different scales. Our work is not the same as Tan et al. in the method and the contents.

Referee's comment:
2. The authors concluded that "we predict the Asian monsoon is strengthening gradually and the Asian monsoon rainfall is increasing gradually in the next several decades or even the next 200 years, in ~ AD 2180±30 the local climate will reach to the next wettest period". This is one of the main conclusion of the paper. However, neither it was shown in Figure 4, nor it was detailed discussed in the paper.

Authors' Response:
Thanks for your comment. The following sentence will been added to the end of the Page 6:

In Figure 4, the curve has reached the top in AD2001, and began to show a downward trend. According to the change rule of the curve, it may be a possible trend that that $\delta^{18}O_R$ will become smaller and smaller in future decades, even in future 200 years, and maybe reach to the lowest in ~ AD 2180±30.

Referee's comment:
3. In page 6, the drought periods deduced from EEMD result are quite similar with Tan et al.' result (Tan et al., 2011a). For example, the drought occurred in ~1350 AD and 1610-1650 AD were clearly shown in the abstract of Tan et al. (2011a). It is understandable, because they use the data of Tan et al. (2011a).

Authors' Response:
In page 6, our results are quite similar with Tan et al.' result (Tan et al., 2011a). Our results from EEMD partly verify Tan et al.' results. Meantime there are still some differences between the two.

Referee's comment:
4. They can't deduce 718yr and 818yr cycles from a 1000-years long record.

Authors' Response:

Yes, I can understand your confusion. But this situation possibly occur in the EEMD. The 718yr and 818yr cycles are calculated from IMF6 and IMF7 components. The cycles of IMF6 and IMF7 components can be approximately shown in Figure 2.

[Figure]

**Fig. 2.** IMF components and the residue of $\delta^{18}O_R$

Referee's comment

5. I think the English of the paper need to be polished further. In addition, I also have some special comments: 1. Page4, line27-29: the speleothem $\delta 18O$ cant reflect rainfall amount in northern China(Zhang et al., 2008; Tan et al., 2011a; Tan et al., 2011b). ), but not in the region of Dongge Cave (Wang et al., 2005; Dykoski et al., 2005). Wang et al. and Dykoski et al. didn't claim that. 2. Page9, line 17: "Ansian monsoon" should be "Asian monsoon".

Authors' Response:

Yes, the English of our paper need to be polished further.

Wang et al. said "Our previous studies have shown that shifts in the oxygen isotope ratio ($\delta^{18}O$) of the stalagmite from the cave largely reflect changes in $\delta^{18}O$ values of meteoric precipitation at the site, which in turn relate to changes in the amount of precipitation and thus characterize the AM strength." in Page854, right column, line19-25 (Science, 308, 2005). But Dykoski et al. didn't explicitly claim that. The expression in Page4, line27-29 will be:

The oxgyen isotope ratio ($\delta^{18}O$) of the stalagmite from cave can reflect change of $\delta^{18}O$ of meteoric precipitation at the site, which in turn relate to the amount of precipitation (Wang et al., 2005; Zhang et al., 2008; Tan et al., 2011a; Tan et al., 2011b).

Thanks for the correction. Page9, line 17: "Ansian monsoon" should be "Asian monsoon".

---

## Author Comment (AC2) · 31 May 2016

Authors' Response to Anonymous Referee #2

We thank the referee for his (her) valuable comments and suggestions. Following is our response.

Referee's comment:
Major comments: The methodology part (EEMD) is a repeat of already published material (Wu and Huang), but the presentation made by the authors is quite obscure, with ill-defined notations and no real effort at synthesizing the methodology and expressing its crucial elements. The authors mainly repeat the description of the Huangye stalagmite data from the original publication, but they do not really discuss (or seem to doubt) how the isotopic record is relevant for their conclusions. The papers of Tan et al. mention correlations between $\delta^{18}O$ and precipitation that are lower than 0.35 (from what I see in the data, this correlation is due to the trend of the last decades). Therefore, a lot of care should be used in interpreting and extrapolating such data.

Authors' Response:
Thanks for your comment. Yes, our presentation about EEMD is short. If the article can be accepted, we will make a more detailed description to the EEMD method.

Tan et al. (Palaeogeography, Palaeoclimatology, Palaeoecology 280, 432-439, 2009) said: "There is a significant negative correlation (R=−0.385, N=33, P<0.05) between the $\delta^{18}O$ of DY-1 and the annual rainfall."
This is a special case of DaYu Cave and Tan et al. analysed the possible reasons.
We cannot find the other correlations (mentioned by Tan et al.) between $\delta^{18}O$ and precipitation that are close to or lower than 0.35.

About Huangye Cave, Tan et al. (Holocene 21, 287-296, 2011) said in abstract: "We developed a composite oxygen isotopic record of cave calcite for the last 1860 years based on three stalagmites from the Huangye Cave in eastern Gansu Province, northern China. The $\delta^{18}O$ values reflect monsoon precipitation changes, with lower $\delta^{18}O$ values representing higher precipitation and vice versa."

Referee's comment:
There is no real discussion of what the cosmogenic isotope record means. The record used by the authors is a mix of several proxies, with different resolutions and temporal coverage. Therefore, the EEMD analysis might only tell something on the way the record was produced, and nothing about solar variability.

Authors' Response:
Thanks for your comment. Yes, the record used by us is a mix of several proxies, with different resolutions and temporal coverage. However the synthesized $^{10}Be$ record can be regarded as a proxy of solar output. We also tried to seach for other proxy such as $^{14}C$ for solar activity in the last 1000 years. But we can not find suitable record with a high resolution.

Referee's comment:

I am a bit surprised that the Fourier analysis of the authors does not find the same peaks as those found by Tan et al. (Palaeogeography, Palaeoclimatology, Palaeoecology 280, 432–439, 2009), with a very similar dataset and the same program for spectral analysis. Since a large part of the interpretation of data relies on those spectra, their instability casts some doubts on the results.

Authors' Response:

Thanks for your comment. Tan et al. (Palaeogeography, Palaeoclimatology, Palaeoecology 280, 432–439, 2009) analysed the periodicities of the $\delta^{18}O$ record of DY-1 ( fome DaYu Cave) by the power spectrum analysis, but they had not analysed the periodicities of the Huangye stalagmite $\delta^{18}O$ data.

Referee's comment:

The spectral analysis results for the three records are very different. The authors seem to manipulate data until they find something that seems coherent. I do not approve of such a procedure.

Authors' Response:

Thanks for your comment. In this paper, the cycles of the three records are mainly shown by EEMD. And these cycles are partly verified by the spectrum analysis results. We donot compare the spectrum analysis results of the three records.

Referee's comment:

In order to be complete, the authors should do the same exercise with volcanic activity, aerosol forcings, etc. The attribution exercise that is done here is very partial.

Authors' Response:

Thanks for your comment. Perhaps, we will further discuss the impact of volcanic activity, aerosol forcings, ENSO, greenhouse gas forcings, etc on the Asian monsoon in the later study. But this is not the focus in the article.

Referee's comment:

The authors make a prediction (p. 9, l. 7): "Meantime we predict the Asian monsoon is strengthening gradually and the Asian monsoon rainfall is increasing gradually in the next several decades or even the next 200 years, in ∼ AD 2180±30 the local climate will reach to the next wettest period." This cannot be serious. The data do not extend into the 21st century. Looking at the time series, I cannot see any hint temperature increases in the 21st century. And, this is were the correlation of 0.3 between precipitation and d18O might mean that precipitation reconstruction fails.

Authors' Response:

Thanks for your comment. The comment is similar to the comment of Anonymous Referee #1. The following sentence will been added to the end of the Page 6:

In Figure 4, the curve has reached the top in AD2001, and began to show a downward trend. According to the change rule of the curve, it may be a possible trend that that $\delta^{18}O_R$ will become smaller and smaller in future decades, even in future 200 years, and maybe reach to the lowest in ~ AD 2180±30.

Referee's comment:

Minor comment: The English is very poor. The manuscript would have needed a cross check from a native English speaker.

Authors' Response:

Yes, the English of the paper need to be polished further.

Referee's comment:

I do not see how EEMD would give better or more insightful results than a wavelet decomposition (the ups and downs of the analysed signal are very symmetric).

Authors' Response:

The wavelet analysis has been widely applied in climate changes, seismic exploration, turbulence, radar monitoring etc in the past 30 years. It has good ability to make multi-resolution analysis in both time domain and frequency domain. Professor Lin Zhenshan (one of the authors) began to adopt wavelet analysis in the study of the atmospheric science twenty years ago (Lin Zhenshan: The wavelet and the hierarchy of climatic system, Met. & Atmo. Phys., 61, 19-26, 1996). Professor Lin has made some contribution to the application and promotion of wavelet analysis in China (Lin Zhenshan and Deng Ziwang: Research on Climate Diagnosis by wavelet analysis, Meteorology Press, Beijing, 1999). However, there are some important limitations as the following:

1) The choice of wavelet basis functions limits the applicability of the technique, as the basic functions of wavelet transformation are fixed and do not necessarily match the shape of the considered data series in every instant in time.

2) The periods deduced from wavelet analysis are often dependent on the empirical parameters. The parameters are dependent on both the methods and experiences by the authors.

3) There is boundary effect the wavelet analysis cannot avoid, which leads to the distortion of the period. Professor Lin has put forward a variety of methods to eliminate the wavelet boundary effect, one of which is the symmetric extension method that is widely adopted in China. Take the boundary line as the symmetry axis, and then copy the original sequence by using the symmetric axis as a mirror in the symmetric extension method. However, it is clear that any elimination of the wavelet boundary effect has its limitation, especially when there are not enough data.

The data in this paper are not long enough to overcome the false information caused by the boundary effect of wavelet analysis. While there are many advantages of EEMD method as pointed out in our paper, we think it is more suitable in this case.

Referee:
The authors should be aware that Tom Crowley passed away in May 2014. Thanking him in the acknowledgments is rather strange.

We were shocked at death of Thomas J. Crowley, and we are very sorry and sad.